# Biomechanical performance of resin composite on dental tissue restoration: A finite element analysis

Abdelhak Ouldyerou[1], Hassan Mehboob[2]*, Ali Mehboob[3], Ali Merdji[1], Laid Aminallah[1], Osama M. Mukdadi[4], Imad Barsoum[3,5]*, Harri Junaedi[2]

1 Department of Mechanical Engineering, Faculty of Science and Technology, University of Mascara, Mascara, Algeria, 2 Department of Engineering Management, College of Engineering, Prince Sultan University, Riyadh, Saudi Arabia, 3 Advanced Digital & Additive Manufacturing Center, Khalifa University of Science and Technology, Abu Dhabi, United Arab Emirates, 4 Department of Mechanical and Aerospace Engineering, West Virginia University, Morgantown, West Virginia, United States of America, 5 Department of Engineering Mechanics, Royal Institute of Technology – KTH, Teknikringen, Stockholm, Sweden

* hassandonate@gmail.com, hmehboob@psu.edu.sa (HM); imad.barsoum@ku.ac.ae (IB)

**Data Availability Statement:** All relevant data are within the manuscript and Supporting information files.

## Abstract

This study investigates the biomechanical performance of various dental materials when filled in different cavity designs and their effects on surrounding dental tissues. Finite element models of three infected teeth with different cavity designs, Class I (occlusal), Class II mesial-occlusal (MO), and Class II mesio-occluso-distal (MOD) were constructed. These cavities were filled with amalgam, composites (Young's moduli of 10, 14, 18, 22, and 26 GPa), and glass carbomer cement (GCC). An occlusal load of 600 N was distributed on the top surface of the teeth to carry out simulations. The findings revealed that von Mises stress was higher in GCC material, with cavity Class I (46.01 MPa in the enamel, 23.61 MPa in the dentin), and for cavity Class II MO von Mises stress was 43.64 MPa, 39.18 MPa in enamel and dentin respectively, while in case of cavity Class II MOD von Mises stress was 44.67 MPa in enamel, 27.5 in the dentin. The results showed that higher stresses were generated in the non-restored tooth compared to the restored one, and increasing Young's modulus of restorative composite material decreases stresses in enamel and dentin. The use of composite material showed excellent performance which can be a good viable option for restorative material compared to other restorative materials.

## Introduction

Dental cavities also known as tooth decay are one of the most common dental complications which are distributed among people of every age [1]. Several factors lead to an ecological imbalance between the inorganic components of dental hard tissues and biofilms, including the physical characteristics of the teeth, environmental conditions, and a sugary diet that contribute to creating tooth cavities [2].

However, modern dentistry prioritizes minimal intervention by emphasizing preventative and non-surgical approaches to safeguard tooth hard tissues before the formation of dental

**Funding:** The authors received no specific funding for this work.

**Competing interests:** The authors have declared that no competing interests exist.

cavities. In this regard, the resin infiltration technique is widely used which involves the application of a low-viscosity resin to infiltrate porous and demineralized inter-crystalline spaces within enamel lesions [3]. In an ex vivo study [4], researchers investigated the treatment of non-cavitated and micro-cavitated caries using a modified resin incorporating silver nanoparticles (AgNP) for both internal and external infiltration, aiming to confer antimicrobial properties. The findings revealed that the AgNP-infused resin effectively hindered the regrowth of microbial biofilms while preserving the mechanical integrity of the enamel and preventing the recurrence of caries [3]. Timely intervention to address demineralized and porous enamel lesions is critical, as untreated lesions may progress to dental cavities.

According to the G.V. Black classification system, a demineralization lesion situated within the pits and fissures of the occlusal (biting) surface is denoted as a Class I lesion, whereas a lesion found on the proximal surface of a posterior tooth is classified as a Class II lesion [5]. Within the Class II category, there exist various subtypes such as mesial and occlusal surfaces (MO) and mesial, occlusal, and distal surfaces (MOD) cavities [6].

The primary goal of cavity treatment is the long-term restoration of functional teeth without their removal. In untreated tooth cavities, the depth of the cavity lesion may increase, which may affect the pulp and thus may lead to inflammation and necrosis of the pulp and loss of vitality [7]. Finding a good approach to restoring the tooth cavity is the holy grail for dentistry researchers which is a challenge for dentists to redesign the cavity carefully without encroaching on the pulp tissue or removing too much dental tissue while choosing the right restoration material for balance between function and aesthetics.

Restorative dental materials commonly employed for direct placement, amalgam has traditionally held a prominent position, although concerns regarding its mercury content and non-aesthetic appearance have led to diminished usage in various communities [8]. In response to these concerns, novel resin-based composites have been developed, characterized by unique biomechanical properties such as cost-effectiveness, non-toxicity, prolonged durability, and enhanced aesthetic qualities [9].

Glass ionomer cements (GIC) are now used as both a restorative material and a fissure sealant in dentistry. It represents a relatively recent commercial development aimed at intentionally remineralizing damaged caries of enamel and dentin [10]. Recent advancements in material development have given rise to glass carbomer cement (GCC), an innovative variant of GICs, which incorporates nanoscale particles and introduces fluorapatite as an additional filler [11]. The flexural strength of GCC was found to be comparable to that of conventional glass ionomer cement [12], underscoring the advancements in material development. Despite these improvements, GICs are still not generally considered the top choice for restorative materials compared to amalgam and resin composites. Statistics show that composite materials (44%) are more commonly used than amalgam (40.9%), and glass-ionomers (13.4%) [10].

Nowadays, the development of dental composites with excellent properties is of high interest to academic researchers, and numerous efforts are currently underway to enhance the mechanical characteristics of dental composites [13]. Recently, micro-nano material science has contributed to improving the mechanical and biological properties of restorative composite materials by adding various filler contents to the organic resin matrix, such as silica nanoparticles, ceramic (zirconia, alumina), hydroxyapatite, calcium phosphate, and metals, in diverse sizes and shapes [14].

In a study [15], the mechanical performance of dental resin composite (DRC) was improved by incorporating silica colloidal nanoparticles. The results of this study revealed that these composites have better mechanical properties compared to many commercial products, even with lower filler content, using only a single filler. Similarly, short glass fiber-reinforced dental composites were prepared by adding silane coupling agents. The results showed that the

glass fibers improved the strength of the composite [16]. In a different investigation, researchers explored the mechanical properties of resin composites and revealed that Grandio material, a hybrid composite with matrix resin of bisphenol A-glycidyl methacrylate (BisGMA) and triethylene glycol dimethacrylate (TEGDMA), and filler of silanized ceramic and silica particles (63% volume filler), exhibited a higher Young's (22.15 GPa ± 2.68) [17].

In bioengineering, finite element analyses (FEA) have been widely used [18–21]. FEA can give freedom for researchers to design cavities and consider various restorative materials to investigate their biomechanical performances. The results can be revealed from the analysis of these models to investigate displacements, micro-movements, and stress/strain distribution in teeth, implants, or host bones under mastication forces.

The purpose of this study was to investigate the behavior of dental tissues using alternative restorative materials to fill cavities of varying designs. Three classes of cavities were chosen, Class I, Class II MO, and Class II MOD, and three types of restoration material were selected, amalgam, GCC, and silica-reinforced polymer composites.

The first null hypothesis maintains that no notable disparity exists in stress distribution within enamel and dentin between an infected tooth and a restored or healthy tooth. The second null hypothesis contends that the stiffness of restorative materials holds no consequential influence on the stress distribution observed in restored teeth.

## Materials and methods

### Construction of 3D finite element models

To generate three-dimensional geometry models, SolidWorks 2020 (Dassault Systèmes, Vélizy-Villacoublay, France) was utilized. A first molar tooth (tooth number 30 in the right mandible) was modeled using a reference image and freeform feature. This tooth has a typical molar shape (a rectangular shape with pits and fissures on it) characterized by multiple cusps and a broad occlusal surface as shown in Fig 1. The poly surfaces were joined using the software's built-in tools and functions. The enamel part was designed by offsetting 2 mm of the dentin crown from the tooth center. Fig 1(a) displays the 3D model of the intact tooth. Three classes of cavities were created, Class I with 30 mm$^3$, Class II MO with 95.59 mm$^3$, and Class II MOD with 176 mm$^3$.

The splint function and extruded cut feature were used to create the cavities and restorative parts with different sizes and dimensions, as shown in Fig 1(b). The thickness of the cortical bone was modeled as 1.5 mm thick using a thickness feature within SolidWorks after designing surfaces to encompass the internal core of the cancellous bone. The PDL (periodontal ligament) and the pulp were not included in our design because it has been established in the literature that when simulating stress distribution at the crown level (regions of interest), the PDL is often excluded due to its well-documented characteristics, including its low stiffness (around 68.9 MPa) and its thin structure (typically 0.2 mm in thickness). These attributes suggest that the PDL's influence on peak stresses at the tooth crown level is negligible. Therefore, the exclusion of PDL and pulp modeling aligns with established modeling techniques [22]. All models were combined using the Boolean cut feature (Fig 1(c)) and exported in STEP, the standardized CAD file format.

### Properties of materials

BisGMA and TEGDMA are resins that are widely used to fill dental cavities [13]. This resin can be reinforced with inorganic ceramic fillers such as silica (SiO$_2$) nanoparticles [23]. Fig 2 depicts the concept of fabricating resin-reinforced SiO$_2$ for use in dental fillings. Various theoretical models were employed to predict elastic moduli of micro-filled composite materials,

**a) CAD modeling of Tooth**

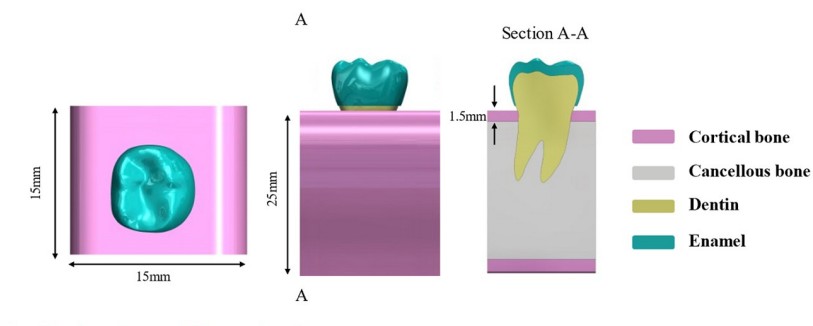

**b) Design of cavities (Infected tooth)**

**Cavity class I**

Depth ≈ 4-4.5 mm
Length = 7.5 mm
Width = 2.2 mm

**Cavity class II MOD (Mesio occlusal distal)**

Depth ≈ 4-4.7 mm
Length = 9 mm
Width = 5 mm

**Cavity class II MO (Mesio occlusal)**

Depth ≈ 4-5 mm
Length = 4 mm
Width = 4 mm

Filling

Amalgam  Composite  GCC

Amalgam  Composite  GCC

Amalgam  Composite  GCC

**c) Assembly**

A

15mm

15mm

25mm

Section A-A

1.5mm

Cortical bone
Cancellous bone
Dentin
Enamel

A

**d) Loading, boundary conditions and mesh**

Uniformly distributed load

600N

Fixed in all directions

Finer elements

Coarse elements

**Fig 1. Models configuration; (a) CAD modeling of the intact tooth, (b) Designs of cavities, (c) Assembly of the models, and (d) Loading, mesh, and boundary condition.**

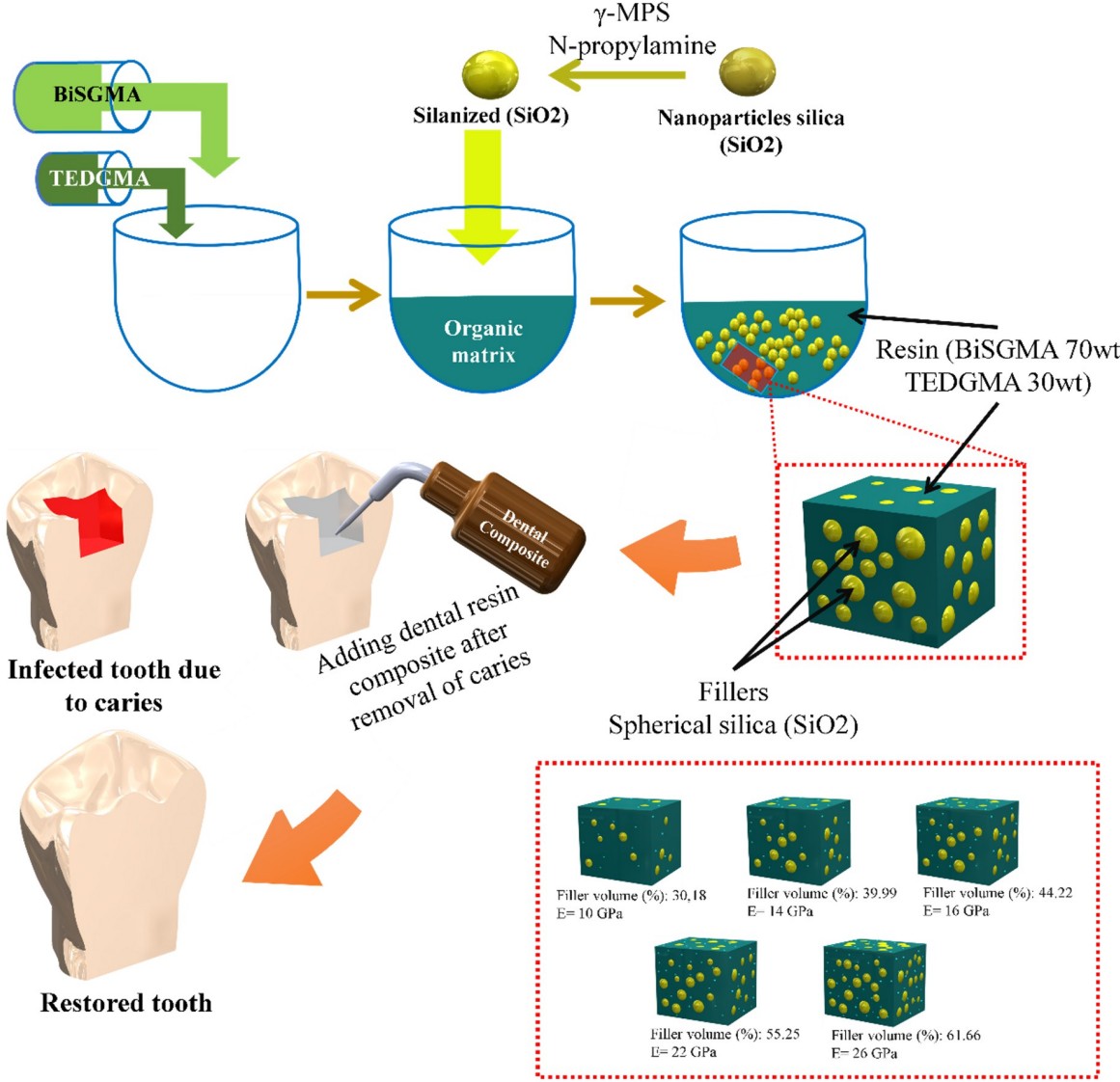

**Fig 2. Representative scheme of the use of SiO$_2$-reinforced resin nanoparticle composites in FEA.**

which were compared with experimental results and showed good agreement [24]. Therefore, this study used these models (Eqs 1–5) to predict the elastic moduli of composites.

Isotropic elastic properties were assumed for the composite fillers and the following formulas were used to calculate the composite Young's modulus:

Einstein model [25]:

$$\frac{E_c}{E_m} = 1 + 2.5V_f \tag{1}$$

Guth model [26]:

$$\frac{E_c}{E_m} = 1 + 2.5V_f + 14.1V_f^2 \tag{2}$$

Counto model [27]:

$$\frac{1}{E_c} = \frac{1 - V_f^{1/2}}{E_m} + \frac{1}{\left[\left(1 - V_f^{1/2}\right)V_f^{1/2}\right]E_m + E_f} \tag{3}$$

Paul model [28]:

$$E_c = E_m \left[\frac{1 - (m-1)V_f^{\frac{2}{3}}}{1 + (m-1)\left(V_f^{\frac{2}{3}} - V_f\right)}\right] \tag{4}$$

Lower bound:

$$E_c = \left[\frac{E_f E_m}{E_f\left(1 - V_f\right) + E_m V_f}\right] \tag{5}$$

Where $E_c$, $E_m$, and $E_f$ are Young's moduli of the composite, resin (BisGMA/TEGDMA) and fillers particles of SiO$_2$, respectively, and $V_f$ is the volume fraction of the reinforcement particles. Fig 3 shows the comparison of Young's modulus obtained from the limited experimental

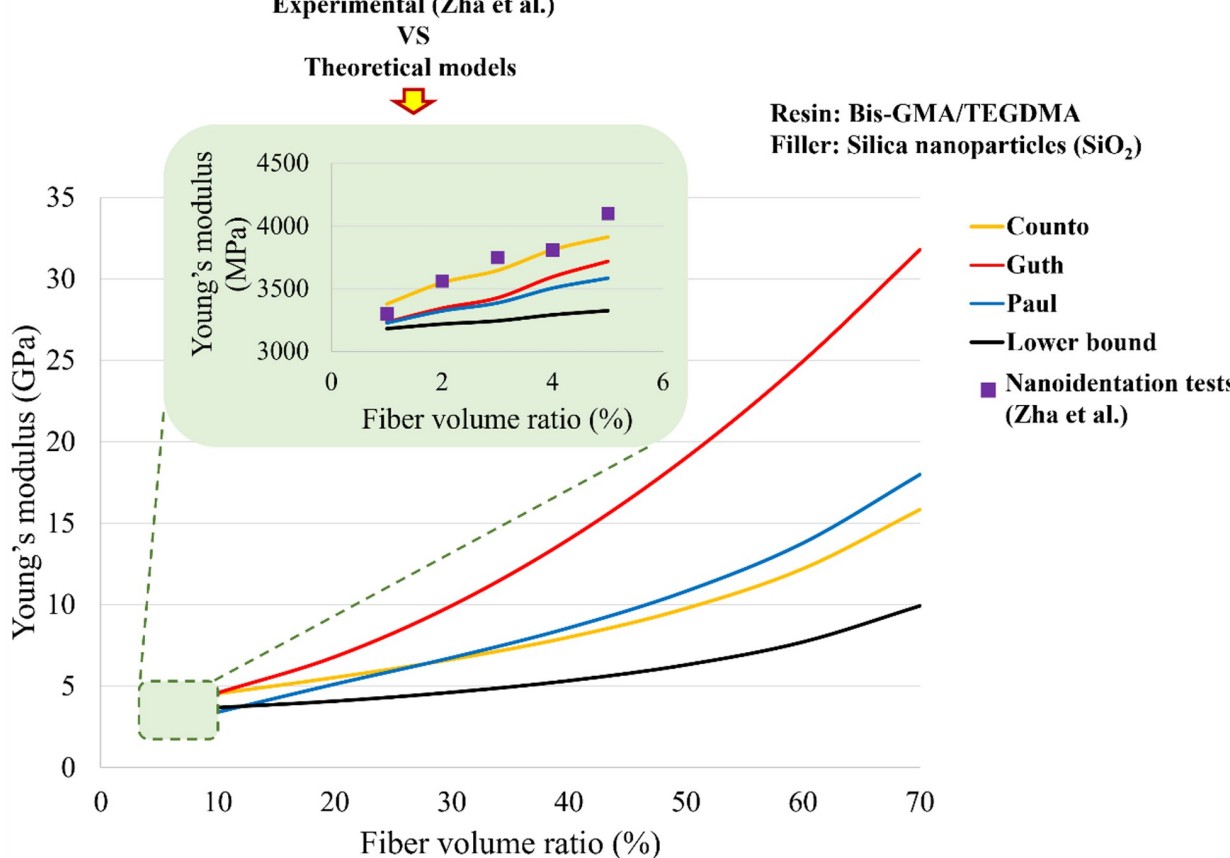

**Fig 3. Comparison of Young's modulus obtained from the experimental [29] and theoretical models for predicting composite Young's modulus with increased fiber volume fraction.**

**Table 1. Properties of the materials used in FEA.**

| Components | Materials | Young's Modulus (GPa) | Poisson's Ratio (v) | References |
|---|---|---|---|---|
| Mandibular Bone | Cortical bone | 13.7 | 0.30 | [30] |
| | Cancellous bone | 1.37 | 0.30 | |
| Molar tooth | Enamel | 84.1 | 0.33 | [31, 32] |
| | Dentin | 18.6 | 0.31 | |
| Resin matrix | Bis-GMA/TEGDMA (30/70 mass ratio) | 3.29 | 0.29 | [33] |
| Reinforcement (filler) | Silica (SiO$_2$) | 73 | 0.17 | |
| Restorative materials | Amalgam | 35.0 | 0.35 | [31] |
| | Composite | 10 | 0.26 | [31] |
| | | 14 | | |
| | | 18 | | |
| | | 22 | | |
| | | 26 | | |
| | GCC (Glass carbomer cement) | 8.3 | 0.30 | [34, 35] |

[29] results with the micromechanics models (theoretical). The analytical models showed satisfactory validation with the experimental results and can hence be confidently used for the estimation of isotropic elastic properties (e.g. Young's modulus) with the assumption that they will follow the same experimental pattern. During the validation process, it was noticed that the Guth model prediction was getting better with the increasing fiber volume fraction while the properties with the Counto model started decreasing. The analytical models were used to calculate Young's modulus of the composite by increasing the volume ratio of the filler (10–70%). Thus, a better Young's modulus of composites was obtained using the Guth model which was used for the comparative study with higher modulus as of amalgam. Fig 3 shows Young's modulus versus particle volume fraction (SiO$_2$) up to 70% fiber volume fraction. Five Young's moduli of the theoretically calculated composite (10, 14, 18, 22, and 26 GPa) were chosen for the simulation. They were labeled C10, C14, C18, C22, and C26, respectively. Amalgam and GCC were also used in finite element simulations as dental material restoration. All materials were assumed to be isotropic, homogenous, and linearly elastic. Young's modulus and Poisson's ratio of all materials are listed in Table 1.

## Mesh, loading, and boundary conditions

For the FEA, the models were imported into ABAQUS software v6.14 (Dassault Systèmes, Vélizy-Villacoublay, France). All the interfaces were considered perfectly bonded as the amalgam is very hard to de-bond as indicated in the literature [36]. The lower surfaces of the bone were fully constrained and a static uniformly distributed occlusal load of 600 N [37] simulating the maximum bite force was applied vertically on the top surface of the crown [38], as shown in Fig 1(d). Each 3D model of the tooth was meshed by finer linear tetrahedral elements (size of 0.05 for the tooth), and the bone was meshed using coarse elements to reduce computational cost. The good quality of the mesh was ensured by keeping the aspect ratio below 5. To make sure that the mesh seed size does not affect the results a mesh convergence was carried out. A series of simulations were conducted with the intact tooth model, varying mesh size (from larger elements with a size of 0.8 to the optimum number of elements with an element size of 0.05). The stress results converged when the total element number yielded more than 500,000 elements (for the intact tooth model, the maximum elements were (512134). The analyses were performed on three parameters, the condition of the molar (intact tooth, infected tooth,

restored tooth), the shape of the cavity (cavities with Class I, Class II MO, Class II MOD), and the dental restorative materials (amalgam, GCC and composites). A total number of 27 simulations were executed.

## Results

Finite element models of the molar crown, in all cases (healthy tooth, infected tooth, restored tooth) were constructed. Various types of dental cavities were also modeled. Conventional and new restorative materials were used in these models. Finite element models were used to simulate the cavities with different restorative materials. Results were extracted from the simulations to estimate the biomechanical performance of dental tissue and restorative material as well.

### Stresses in different cavities

Three cases of cavities were studied, Class I, Class II MO, and Class III MOD, and they differ in volume sizes and designs. Class I cavity was formed in the occlusal crevice with 30 mm$^3$ in volume. Class II MO and Class II MOD were prepared with 95.59 mm$^3$ and 176 mm$^3$, respectively. Fig 4. shows the stress distribution in enamel and dentin, respectively, with cavity Class I.

The numerical simulation findings are reported in Table 2. It was observed that higher von Mises stress was in the infected tooth with a value of 44.40 MPa in the enamel and 35 MPa in the dentin. Same note for maximum principal stress, 48.06 MPa in the enamel, and 40.20 in the dentin. As for the healthy tooth, it can be noted that von Mises stresses were lower compared to the infected tooth, with 21 MPa in the enamel, and 15.32 MPa in the dentin. The same pattern follows for principal stresses.

Fig 5. show von Mises stress distribution in the enamel and dentin with cavity Class II MO. The levels of stress experienced by the diseased tooth were found to be greater when compared to the healthy tooth. Higher von Mises stress was observed in the enamel (54.91 MPa), and 39.18 MPa in the dentin. It was noted that stresses in the case of cavity Class II MO were higher than those obtained with cavity Class I.

Fig 6. depicts the von Mises stress distribution in the enamel and dentin with cavity Class II MOD. When comparing a healthy tooth with an infected tooth with cavity Class II MOD. It is evident that greater levels of stress are present in infected teeth. The maximum stress in the enamel of the infected tooth was 170.80 MPa, and 72.10 MPa for the dentin. Through this comparison, it could be observed that as the size of the cavity increases, the stress on the enamel and dentin increases. Stresses (von Mises and principal stresses) were higher in the tooth with cavity Class II MOD compared to the tooth with cavity Class I and Class II MO, as well stresses were higher in the enamel than dentin in all cases. Enamel and dentin have a compressive strength of around 384 MPa and 297 MPa, respectively [22]. All minimum principal stress values obtained in enamel and dentin under loading of 600 N were below the compressive strength.

Enamel tensile strength is approximately 42 MPa, while the tensile strength of dentin is 44 MPa [22]. Fig 7. shows a comparison between principle stresses in all models, maximum principal stresses in the dentin were below the tensile strength. The same note is for the enamel, except for the infected tooth with cavity type II MOD, where the maximum stress (120 MPa) was greater than the tensile strength of enamel, and thus failure may occur. The comparison and findings in Table 2 demonstrate how the teeth behave differently in different biomechanical conditions.

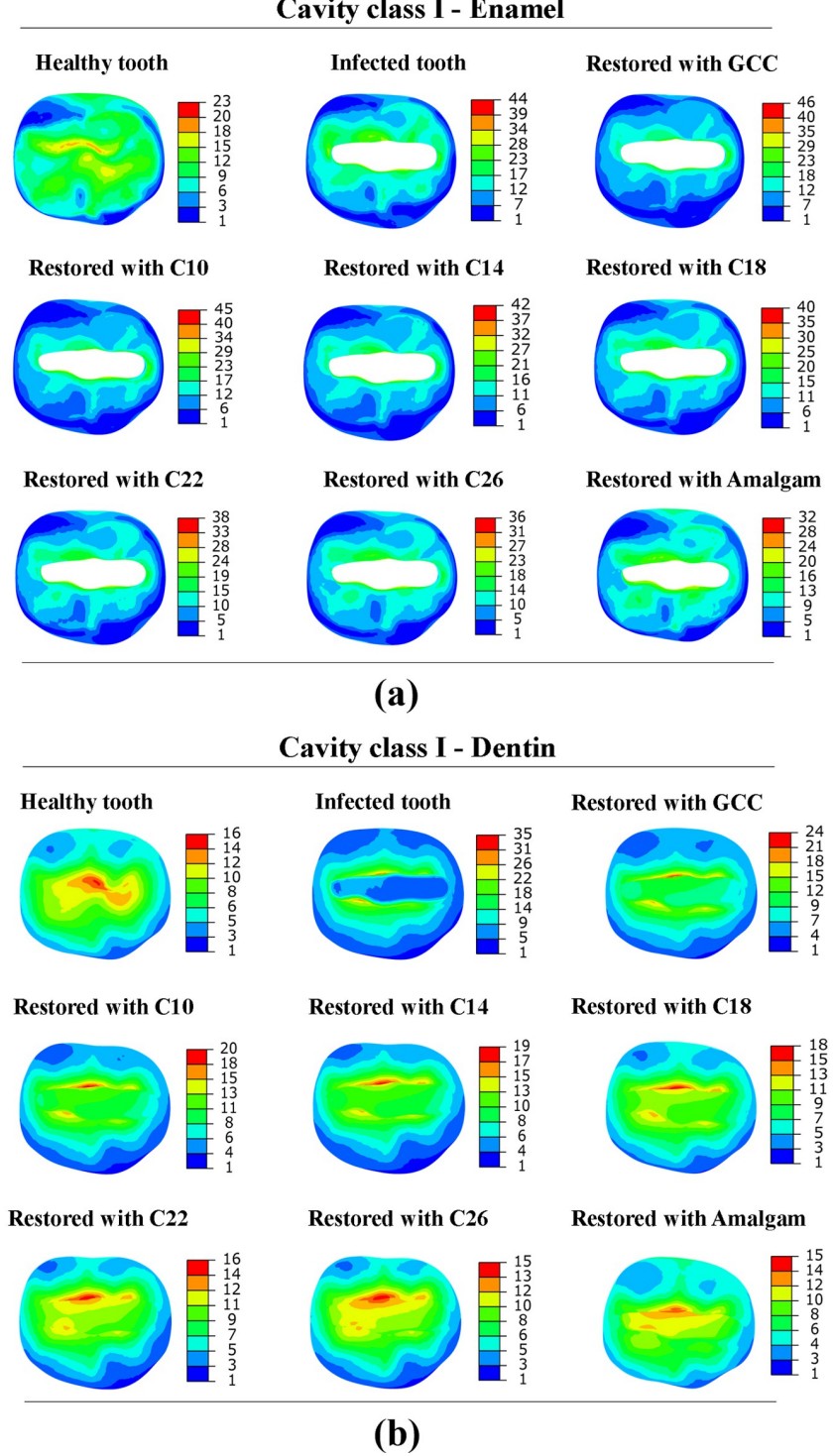

**Fig 4. Von Mises stress distribution in; (a) enamel and (b) dentin tissue with cavity Class I (in MPa).**

**Table 2. Max stresses in enamel, dentin, and restorative parts of cavities (in MPa).**

|  | Max stress | Intact tooth | Infected tooth | GCC | C10 | C14 | C18 | C22 | C26 | Amalgam |
|---|---|---|---|---|---|---|---|---|---|---|
|  |  |  | **Enamel** |  |  |  |  |  |  |  |
| **Cavity Class I** | vM stress | 23.11 | 44.40 | 46.01 | 45.07 | 42.10 | 39.65 | 37.54 | 35.70 | 31.90 |
|  | maxP stress | 13.27 | 20.43 | 17.73 | 17.38 | 16.72 | 16.30 | 16.00 | 15.70 | 15.48 |
|  | minP stress | 24.00 | 48.06 | 50.87 | 49.62 | 46.27 | 43.41 | 40.90 | 38.70 | 34.40 |
| **Cavity Class II MO** | vM stress |  | 54.91 | 43.64 | 42.16 | 39.85 | 38.10 | 37.02 | 36.20 | 34.50 |
|  | maxP stress |  | 25.20 | 15.33 | 14.70 | 14.20 | 13.95 | 13.69 | 13.47 | 13.03 |
|  | minP stress |  | 58.27 | 46.17 | 43.68 | 42.00 | 40.71 | 39.60 | 38.66 | 37.30 |
| **Cavity Class II MOD** | vM stress |  | 170.80 | 44.67 | 40.40 | 34.26 | 30.03 | 26.92 | 24.50 | 20.41 |
|  | maxP stress |  | 120.00 | 27.66 | 23.12 | 18.40 | 15.36 | 13.24 | 11.68 | 10.38 |
|  | minP stress |  | 202.25 | 41.06 | 39.13 | 34.20 | 30.52 | 27.53 | 25.07 | 20.24 |
|  |  |  | **Dentin** |  |  |  |  |  |  |  |
| **Cavity Class I** | vM stress | 15.32 | 35.04 | 23.61 | 22.41 | 19.49 | 17.50 | 16.18 | 15.20 | 14.03 |
|  | maxP stress | 1.97 | 7.87 | 2.68 | 2.60 | 2.55 | 2.48 | 2.43 | 2.37 | 2.34 |
|  | minP stress | 18.57 | 40.20 | 26.25 | 24.49 | 21.24 | 19.10 | 17.44 | 16.30 | 15.10 |
| **Cavity Class II MO** | vM stress |  | 45.19 | 39.18 | 37.89 | 35.98 | 34.64 | 33.63 | 32.87 | 31.28 |
|  | maxP stress |  | 24.09 | 6.54 | 5.77 | 4.57 | 3.91 | 3.80 | 3.72 | 3.54 |
|  | minP stress |  | 46.84 | 45.91 | 44.90 | 42.86 | 41.39 | 40.28 | 39.41 | 37.60 |
| **Cavity Class II MOD** | vM stress |  | 72.10 | 27.50 | 23.83 | 19.61 | 16.61 | 14.38 | 13.51 | 13.70 |
|  | maxP stress |  | 28.04 | 7.37 | 6.11 | 4.78 | 4.06 | 3.54 | 3.14 | 2.76 |
|  | minP stress |  | 81.00 | 22.65 | 21.00 | 17.24 | 14.56 | 14.07 | 14.30 | 14.66 |
|  |  |  | **Restorative part** |  |  |  |  |  |  |  |
| **Cavity Class I** | vM stress |  |  | 10.90 | 12.50 | 14.96 | 16.88 | 18.45 | 19.79 | 21.34 |
|  | minP stress |  |  | 12.61 | 13.57 | 15.95 | 17.87 | 19.27 | 20.53 | 23.40 |
| **Cavity Class II MO** | vM stress |  |  | 16.21 | 16.40 | 16.70 | 17.27 | 19.28 | 21.14 | 23.96 |
|  | minP stress |  |  | 16.25 | 17.26 | 19.17 | 20.46 | 22.47 | 24.56 | 30.00 |
| **Cavity Class II MOD** | vM stress |  |  | 13.77 | 14.00 | 14.08 | 13.94 | 15.54 | 15.66 | 16.93 |
|  | minP stress |  |  | 18.40 | 18.41 | 18.71 | 18.86 | 18.90 | 18.99 | 19.58 |

vM*: Von Mises stress,

maxP*: Maximum principal stress,

minP *: Minimum principal stress in MPa

## Effect of dental restoration material

The tooth models with three classes of cavities were restored with 7 different materials (GCC, composite C10-C26, amalgam). The stress distribution in the restored tooth with different cavities and restorative materials is shown in Figs 4–6. The peak von Mises stress in different cavities was at the interfaces between the restorative part and the enamel, as well as between the restorative part and the dentin. The FEA findings (reported in Table 2) revealed that von Mises stress was higher in GCC material, with cavity Class I (46.01 MPa in the enamel, 23.61 MPa in the dentin), and for cavity Class II MO von Mises stress was 43.64 MPa, 39.18 MPa in enamel and dentin respectively, while in case of cavity Class II MOD von Mises stress was 44.67 MPa in enamel, 27.5 in the dentin. Through simulation results, it was noted that lower stresses were in the amalgam-restored tooth for cavity Class I, 31.90 MPa in the enamel, and 14.03 MPa in the dentin. The results indicated that the highest Young's modulus of the restorative material was associated with the lowest peak stresses observed at the enamel and dentin levels. This highlights the crucial role played by Young's modulus of the restorative material in

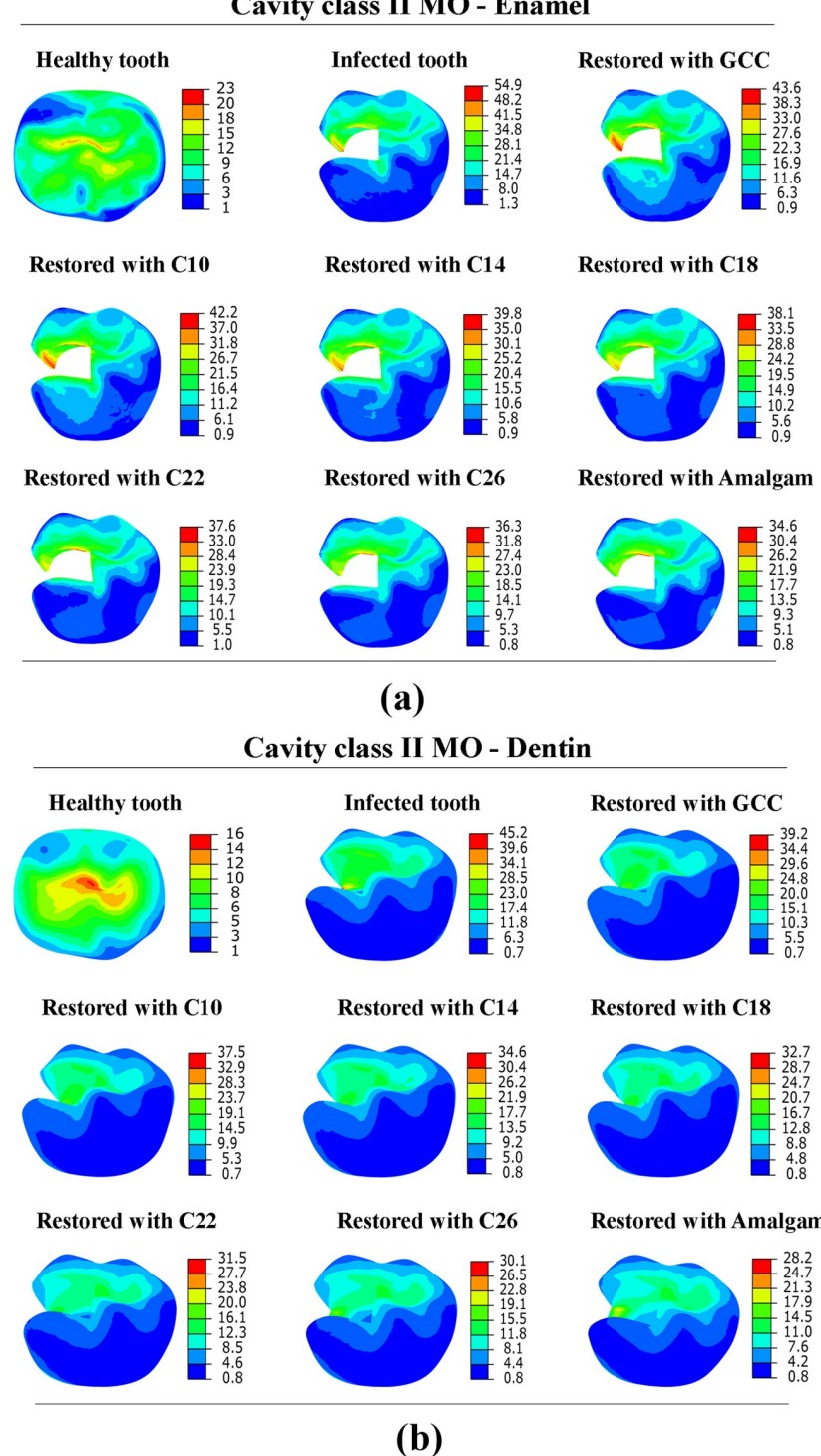

**Fig 5. Von Mises stress distribution in; (a) enamel and (b) dentin tissue with cavity Class II MO (in MPa).**

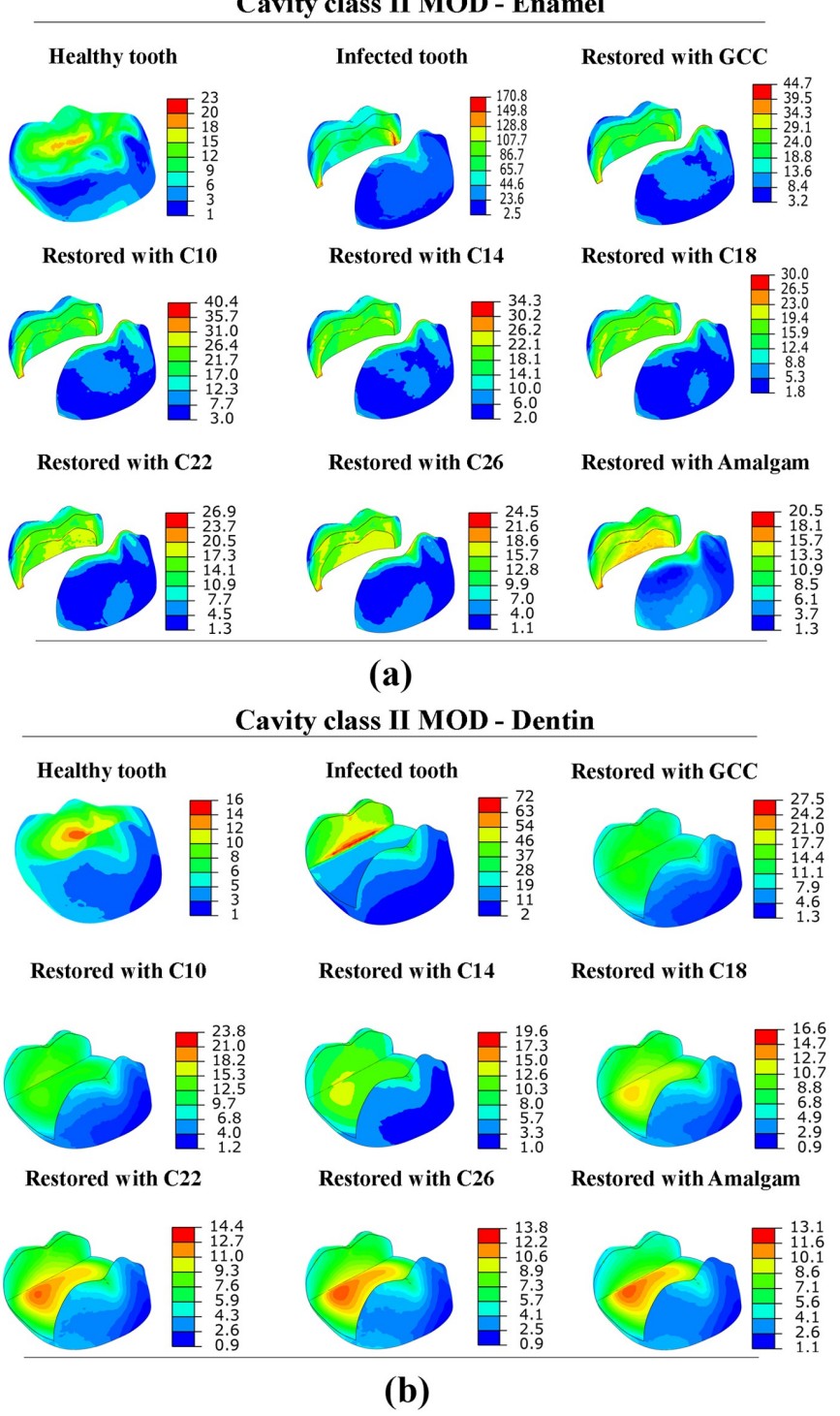

**Fig 6. Von Mises stress distribution in; (a) enamel and (b) dentin tissue with cavity Class II MOD (in MPa).**

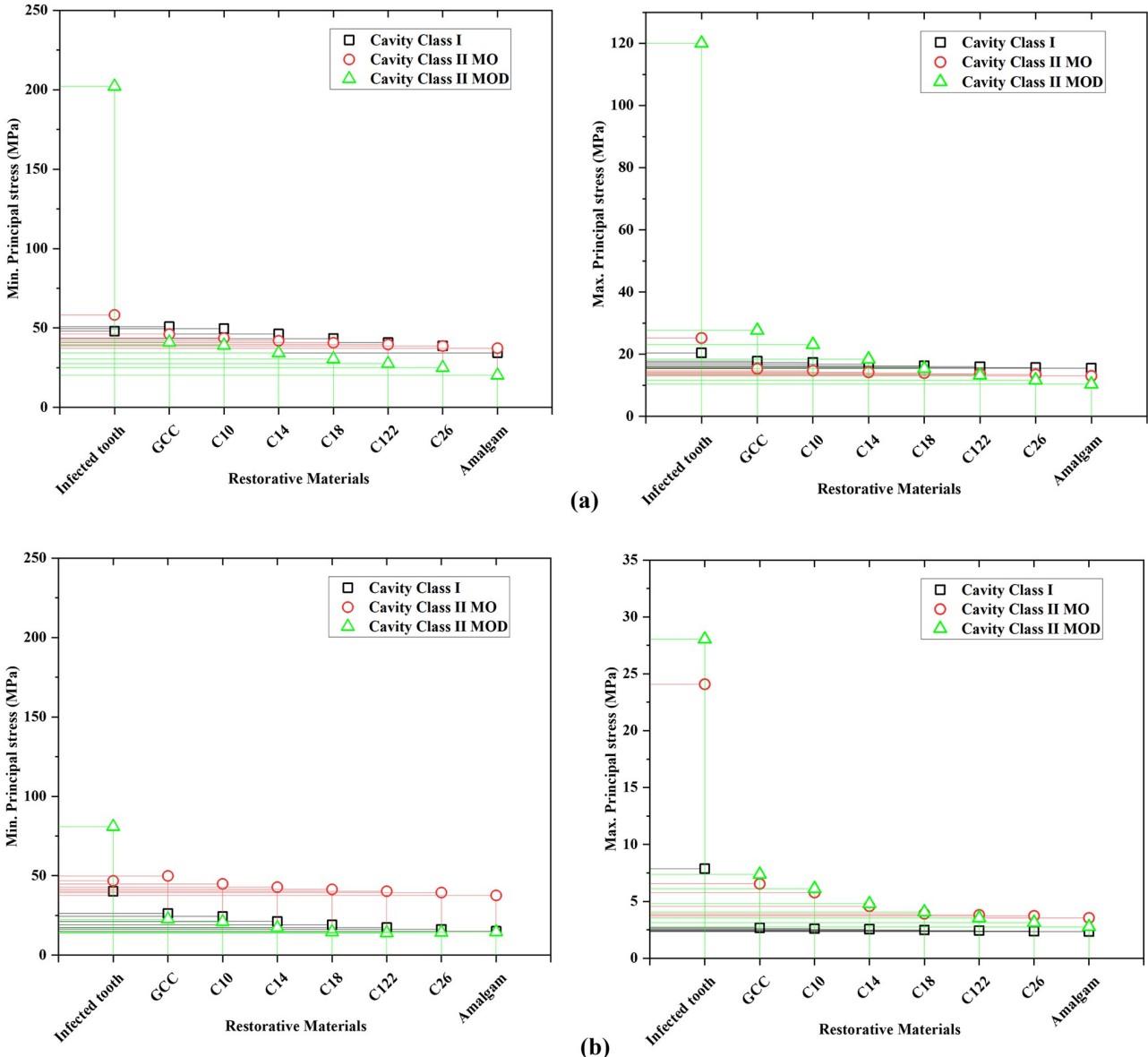

**Fig 7. Comparison of principal stress in the restored tooth with different cavities and different restorative materials; (a) enamel, and (b) dentine.**

determining the stress experienced by enamel and dentin tissues. In contrast, with the composite materials with different Young's moduli (26, 22, 18, 14, 10 GPa), we noticed that stresses were higher with the lowest Young's modulus (10 GPa), as we increased Young's modulus (26 GPa), the stress was decreased in the enamel and dentin. Additionally, it is worth mentioning that all stress levels observed in the restored tooth were lower in comparison to the tooth affected by the cavity. As for the restorative part, it was remarked that increasing the effective stiffness of restorative material increased the stress at the level of the restorative part, von Mises stress was lower in GCC restorative part (10.9 MPa), and it was higher in the amalgam restorative material (21.34 MPa), and this was in the case of cavity Class I. Fig 7. shows the comparison of principal stress in enamel (Fig 7(a)) and dentin tissue (Fig 7(b)). All principal

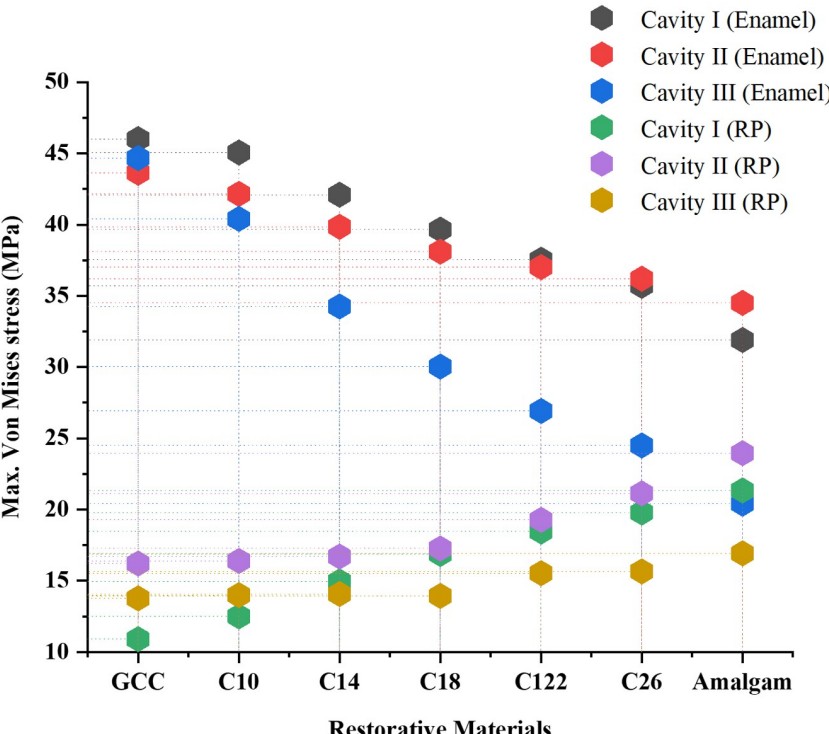

**Fig 8. Maximum von Mises stress in the enamel and restorative material with different cavities.**

stresses (PS) values reported in restored cavities in enamel and dentin were lower than those of non-restored cases (Infected teeth). Fig 8. shows the maximum von Mises stress in the enamel and restorative part. The greater the stiffness of the restoration material, the lower the *stress* at the enamel level, and vice versa for the restorative part. The higher the stiffness, the higher the stress at the restorative material level.

## Discussion

This study examines three different restorative materials (Amlgam, GCC, Composite) in relation to three categories of dental cavities, specifically Class I and Class II cavities, which include MO and MOD types. Teeth lacking any form of restoration are designated as infected teeth, while teeth with restorations are categorized as restored teeth. In the field of restorative dentistry, understanding the behavior of restored teeth has proven to be a multifaceted challenge. This complexity arises from the wide array of restorative materials available, encompassing both traditional and advanced options as each material type can exhibit distinct properties and behaviors. These advancements have the potential to overcome the limitations of amalgam material, including their ability to control mineral loss and modulate biofilm formation. Regarding GIC, limited evidence [39] suggests that restorations made with this material exhibit a notably superior preventive effect against the development of new caries when compared to other restoration materials.

The results revealed that the first null hypothesis, suggesting no significant difference in stress distribution between infected, healthy, and restored teeth, was not supported and was consequently rejected. The findings indicated that stress levels were notably higher in infected teeth when compared to both restored and healthy teeth. This difference in stress distribution

can be attributed to variations in the load application area within the cases under investigation. Specifically, infected teeth exhibited a reduced occlusal area subjected to a 600 N force in comparison to their healthy counterparts. It is important to note that the intensity of a distributed load is contingent upon the force per unit area. Therefore, teeth with cavities transmit a heightened level of stress to the dentin tissue. Consequently, the reduction in tooth structure significantly diminishes tooth stiffness [40]. The findings revealed that as more tooth material was removed, an increased concentration of stresses was observed within the enamel and dentin tissues.

The maximum principal stress in an infected tooth's enamel was 201 MPa, which is significantly higher than the ultimate tensile strength of enamel at approximately 42 MPa [41]. This means that the enamel is subjected to stress levels that are well beyond its capacity to withstand without exhibiting permanent deformation or damage. Moreover, in various conditions like tooth decay, consumption of acidic foods and drinks or chronic acid reflux may result in cracks, fractures, and even complete failure of the tooth structure [42].

Addressing early detection and treatment of dental infections promptly becomes pivotal in preventing the progression of damage and the subsequent necessity for extensive restorative techniques. Research investigating resin infiltration techniques [3, 43, 44] substantiates this notion, demonstrating their capacity to halt the advancement of white spot lesions. This underscores the importance of timely intervention as a preventive measure against the need for extensive dental restorations.

Additionally, the principal stresses were higher in the infected tooth with Class II MOD (Mesio-Occlusal-Distal) cavity compared to other cavities. Class II MOD cavity preparations involve the removal of a significant portion of the tooth structure, including both the mesial and distal surfaces. This type of preparation is more extensive than Class II MO and Class I Occlusal preparations, which involve the removal of tooth structure from a single surface. The removal of more tooth structure in Class II MOD preparations results in a larger cavity volume and a greater loss of tooth structure, which can lead to increased stresses in the remaining tooth structure. This emphasizes the need for careful consideration and selection of dental materials used for cavity II MOD dental restorations to ensure they can withstand the stresses imposed on them during function. Furthermore, the findings align with previous research indicating that a decayed tooth can generate greater stress than an undamaged tooth [45]. Nowadays, researchers are looking for means to standardize cavity preparations so that the underlying tooth can be preserved.

The second null hypothesis, which stated that the stiffness of restorative materials has no consequential influence on the stress distribution observed in restored teeth, was also not supported and therefore was rejected. The findings indicate that the type of cavity and the varying stiffness of restorative materials do indeed affect the distribution of stress within the enamel and dentin tissues of restored teeth.

Higher stress in enamel and dentin was in the restored tooth with GCC material and the lower stress was in the restored tooth with amalgam material. In the case of composite (C) material, the stress in enamel and dentin was less when Young's modulus of composite material was lower (10 GPa), and vice versa. Results are in agreement with a previous study [31], which has reported that for a certain Young's modulus of the restoration material, the first principal stress becomes minimum at the enamel.

When it comes to the stiffness of restoration materials, a study [46] revealed that a higher Young's modulus of the restorative substance leads to a reduction in stress on both enamel and dentin, even when specifically discussing amalgam. A study [47] demonstrated that restoration with various cavity designs (five distinct Class II restored with composite resin) resulted in different stress magnitudes during the processes of polymerization shrinkage and loading

effects on dental tissue. This underscores the influence of cavity geometry on stress distribution.

According to another study [48], endocrown design preparation has a significant impact on long-term success, and restorative materials with a lower modulus of elasticity had more homogeneous stress distribution on the dentin tissue, whereas materials with high young modulus had low stress transmission to the dentin tissue. Another study [35] reported that the GCC material transfers more von Mises stress in the dentin tissue compared to amalgam and composite materials with Class II MOD cavities.

In conjunction with the aforementioned findings regarding the influence of different cavities and restorative materials on stress distribution in teeth, a study [49] explored the impact of restorative material choice on stress distribution within endocrown restorations. The research revealed that the selection of restorative material had a notable effect on stress distribution patterns within endocrown restorations, while the application of fibers did not significantly alter stress distribution in either endocrown group.

The mechanical properties of restorative materials, particularly Young's modulus, are crucial in determining the clinical performance of endodontically treated teeth that have been restored. In this context, the stiffness of restorative materials is intentionally kept lower than that of enamel to strengthen the tooth structure rather than weaken it.

Composite resins, designed to replicate the natural appearance of teeth and closely mimic their mechanical properties, demonstrate a favorable attribute in adapting to the natural flexure and movement of teeth during function. This adaptability potentially contributes to a more uniform distribution of stress forces, mitigating the likelihood of stress concentration points at the adhesive-tooth interface. An in vitro study corroborated this behavior, demonstrating that the use of composite resin reduces stress concentration at the interface between the adhesive and the tooth compared to materials with a higher Young's modulus than enamel [42].

Results showed less variation in the peak stresses between the tooth restored with amalgam and the tooth with composite C26 (which has a Young's modulus of 26 GPa) when compared to other composites with lower Young's moduli. Biomechanically, there was no statistically significant difference between amalgam and composite in terms of fractures, but the longevity of the composite in posterior teeth remains less. Despite other options being available, composite has emerged as the primary material preferred by dentists for restorative purposes, with favorable attitudes being expressed towards its use [50]. Higher stresses were observed in the enamel compared to the dentin tissue. This difference can be attributed to the fact that enamel is considered to be harder than dentin and possesses higher wear resistance. These two tissues also differ in composition and internal structures and have distinct biomechanical roles [51]. Additionally, in comparison to GCC, composite resins are more effective [52], and they release less fluoride. According to some authors [53, 54], composite resin materials are favored over glass ionomer cement for sealing pits and fissures due to their superior retention properties.

Dentists generally focus on the necessity of preserving dental tissue [55] over quantifying the number of surgical restorative procedures performed. Achieving this goal requires the development of new tools, reimbursement incentives, and collaboration to coordinate the implementation of an improved dental cavity management system across various domains [56].

With the increasing use of digital impressions and 3D printing in dental clinics [57], it is now possible to leverage CAD-CAM techniques to produce nanohybrid restorative materials that are better suited for dental tissue. Furthermore, recent developments in artificial neural network models have led to the creation of predictive tools for post-streptococcus mutants in dental cavities, which can be integrated with digital impressions and 3D printing technologies to enable even more accurate and personalized treatment options for dental patients [58].

FEA stands as a potent tool in the realms of bioengineering and dentistry, and our study's primary objective was to scrutinize the repercussions of dental cavities and different restorative materials on the performance of enamel and dentin tissues.

It is essential, however, to acknowledge certain limitations inherent to our research. Notably, we employed a simplified assumption by considering all materials as homogenous, isotropic, and linearly elastic. Furthermore, our loading conditions were somewhat idealized, and variations such as oblique or parafunctional loading could impose varying stress patterns on the tooth structure. In this study, finite element models did not include PDL considering that the exclusion of PDL affects minimal to the stresses in the crown and surrounding tissues [22]. Another facet of consideration is the derivation of Young's modulus for the composite material, which was based on theoretical models rather than empirical testing. Although these models have been substantiated in previous research, the importance of experimental verification of the calculated Young's modulus cannot be overstated. Hence, prospective studies should prioritize the conduction of such experiments to corroborate the accuracy of the theoretical models employed here, thereby enriching our comprehension of composite material properties.

## Conclusion

From this study, it was concluded that the geometry of cavities and different restorative materials affect the biomechanical performance of the restored tooth. Nanocomposite resin is a good alternative material for dental hard tissue restoration especially for patients who are concerned about the potential toxicity of mercury or aesthetic restoration.

- Infected or damaged teeth with cavities are prone to higher stresses as compared to healthy teeth.

- Larger cavities generate higher stresses in enamel and dentin tissues, thus, to protect the remaining teeth, the larger and smaller cavities should be filled as early as possible.

- Without filling the cavity, the stresses may reach higher than the strength of the enamel which may cause further damage to the tooth.

- The restoration of Class II MOD cavity is very tricky and complex because of the removal of large tissue, which may require more care and maintenance during their normal function.

- The selection of restorative material is the key to efficiently restoring the function of the tooth. Therefore, resin-based composites are good alternative materials due to the tailorable Young's modulus according to the biomechanical and clinical requirements.

- Overall, amalgam and composite C26 (26 GPa) outperformed biomechanically but composites are preferred over amalgam due to aesthetic and clinical concerns of amalgam.

## Supporting information

**S1 Dataset.**
(ZIP)

## Acknowledgments

The authors would like to thank Prince Sultan University for their support.

## Author Contributions

**Conceptualization:** Hassan Mehboob, Laid Aminallah.

**Data curation:** Abdelhak Ouldyerou.

**Formal analysis:** Imad Barsoum.

**Investigation:** Abdelhak Ouldyerou, Imad Barsoum.

**Methodology:** Abdelhak Ouldyerou, Hassan Mehboob.

**Software:** Osama M. Mukdadi.

**Supervision:** Ali Merdji.

**Validation:** Ali Mehboob.

**Visualization:** Ali Merdji, Laid Aminallah, Osama M. Mukdadi.

**Writing – original draft:** Abdelhak Ouldyerou.

**Writing – review & editing:** Hassan Mehboob, Ali Mehboob, Harri Junaedi.

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
