## [Decision Letter · Decision Letter 0]

4 Sep 2023

PONE-D-23-26356Biomechanical performance of resin composite on dental tissue restoration: A computational analysisPLOS ONE

Dear Dr. Mehboob,

Thank you for submitting your manuscript to PLOS ONE. After careful consideration, we feel that it has merit but does not fully meet PLOS ONE’s publication criteria as it currently stands. Therefore, we invite you to submit a revised version of the manuscript that addresses the points raised during the review process.

We look forward to receiving your revised manuscript.

Kind regards,

Artak Heboyan, Ph.D.

Academic Editor

PLOS ONE

Journal Requirements:

3. Thank you for stating the following financial disclosure: "The funders contributed equally in the preparation of the manuscript".

5. Please note that in order to use the direct billing option the corresponding author must be affiliated with the chosen institute. Please either amend your manuscript to change the affiliation or corresponding author, or email us at plosone@plos.org with a request to remove this option.

Please carefully address the reviewers` comments and revise the manuscript extensively.

Reviewers' comments:

Reviewer's Responses to Questions

**Comments to the Author**

1. Is the manuscript technically sound, and do the data support the conclusions?

Reviewer #1: Yes

Reviewer #2: Yes

Reviewer #3: Partly

2. Has the statistical analysis been performed appropriately and rigorously? 

Reviewer #1: N/A

Reviewer #2: N/A

Reviewer #3: I Don't Know

3. Have the authors made all data underlying the findings in their manuscript fully available?

Reviewer #1: No

Reviewer #2: No

Reviewer #3: Yes

4. Is the manuscript presented in an intelligible fashion and written in standard English?

Reviewer #1: Yes

Reviewer #2: No

Reviewer #3: No

5. Review Comments to the Author

Reviewer #1: The present study investigated the biomechanical performance of various dental materials when filled into different cavity designs and their effects on surrounding dental tissues. The subject is intriguing and has the potential to contribute to the scientific literature.

Title:

Please indicate that this is an FEA study.

Abstract:

Loading information is missing.

Introduction:

The introduction lacks an explanation of different cavities and their significance as evaluation factors. Additional information about glass carbomer cement and its comparison with resin composite and amalgam should be introduced. Include a hypothesis at the end of this section.

Methods:

The tooth was modeled using a reference image and freeform features. Provide a description of the tooth number and shape. Explain how cortical bone thickness was defined. Justification for not including the PDL and pulp in the design needs revision. While it's acknowledged that soft tissue could affect stress distribution, the statement claiming no effect should be omitted. During the validation process, it was observed that the Guth model's prediction improved with increasing fiber volume fraction, whereas properties with the Counto model decreased. Provide a justification for this observation. Clarify that Young's Modulus is theoretical. Specify the aspect ratio of your mesh. Explain the choice of focusing on von-Mises stress over the first principal stress.

Discussion:

Enhance the discussion regarding the comparison of different cavity designs. Provide more information about model validation and FEA limitations concerning PDL, soft tissue, and parafunctional loading.

Conclusion:

The conclusion is overly lengthy. Condense the text, remove redundant results, and present only the most clinically significant findings to the readers.

Reviewer #2: Intro

- "Several factors lead to an ecological imbalance between (...) that contribute to creating tooth cavities [2]." Please add some short thoughts on the pre-cavity stage. Remember that this would seem even more (or at least equally) important to prevent progress of caries (and cavitation). Refer to https://www.quintessence-publishing.com/deu/de/article/840164/quintessence-international/2009/08/closing-the-gap-between-oral-hygiene-and-minimally-invasive-dentistry-a-review-on-the-resin-infiltration-technique-of-incipient-proximal-enamel-lesions, and to https://journals.plos.org/plosone/article?id=10.1371/journal.pone.0228249. Discuss adequately.

- Provide digestible units/divide into paragraphs.

- Avoid Authors' names with your full text.

- "The purpose of this study is to investigate (...)." Your study has already been finished, right? Must read "The purpose of this study was to investigate (...)."

Results

- "(...) with cavity class I." must read "(...) with cavity Class I."

- Same with "cavity class II". Revise thoroughly.

Discussion

- "The behavior of restored teeth is not well characterized due to the variety of restorative materials (both traditional and advanced materials)." Meaning would seem unclear, please expand your thoughts, and provide references.

- Same with "the stress was greater in the infected teeth". Please clarify.

- Refer to https://journals.sagepub.com/doi/10.1177/00220345890680111401

when explaining your thoughts.

- Again, provide digestible units by paragraphing.

- Add some thoughts on the limitations of your study.

Conclusion

- "The aim of this study was to analyze how (...)." This is not considered a conclusion from your study.

- Same with "Five restorative materials were studied, glass-carbomer cement, amalgam, and dental composite with stiffness variation (10 G Pa to 26 G Pa with an increment of 4)."

- Same with "Nanocomposite resin may be an excellent choice (...)."

- With your Conclusion section, please stick exclusively to your revised aims. Do not simply repeat your results here. Do not speculate. Instead, provide a reasonable and generalizable extension of your outcome.

Refs

- Revise for uniform Journal style. No capital letters with the various Titles.

In total, this would seem an interesting draft, easily intelligible, and considered ready to proceed after some revisions, please see below.

Reviewer #3: The goal of this study is to examine how dental tissues react when cavities of various shapes are filled with alternative restorative materials. Three classes of cavities were chosen, class I (occlusal), class II MO (mesio-occlusal), class II MOD (mesio-occlusal-distal), and three types of restoration material were selected, amalgam, glass carbomer cement (GCC), and silica-reinforced polymer composites with five different Young’s moduli. While it is good that the results have practical impact in terms of application, there are problems in modeling, especially in the creation of finite element models. Therefore, I think it needs to be revised and needs a second check.

The figures and pictures are so blurry that it is difficult to read.Please enrich the quality of the figures and pictures.

Wouldn't the periodental ligament have an effect on stress changes? Not considered in the modeling.

The Author said that ”Each 3D model of the tooth was meshed by finer linear tetrahedral elements (size of 0.05 for the tooth), and the bone was meshed using coarse elements to reduce computational cost”. However, while finite elements are a way of representing the structure as an elastic structure, the element quality has a very important effect on this representation. Three classes of cavities were used. However, there will be geometry changes. How did the same element size suffice for all of them? Linear element was used and compared to quadratic type, tetrahedron element has poor performance and high margin of error. I suggest the authors to do a convergence analysis and quadratic element.

It would be better for comparison if the fringes used in the figures in the stress distributions have the same range.

I recommend the author to add the following publication to the literature search. In this publication "The restorative material type changed the stress distribution of endocrown restorations. The fiber application did not affect the stress distribution in either endocrown group. But, more stress absorption was observed in fiber under IPS-E than C-E." is the conclusion that will contribute to the content of the publication.

6. PLOS authors have the option to publish the peer review history of their article (what does this mean?). If published, this will include your full peer review and any attached files.

Reviewer #1: No

Reviewer #2: No

Reviewer #3: No

---

## [Author Response · Author response to Decision Letter 0]

28 Oct 2023

Reviewers' comments:

Reviewer's Responses to Questions

Comments to the Author

1. Is the manuscript technically sound, and do the data support the conclusions?

Reviewer #1: Yes

Reviewer #2: Yes

Reviewer #3: Partly

2. Has the statistical analysis been performed appropriately and rigorously?

Reviewer #1: N/A

Reviewer #2: N/A

Reviewer #3: I Don't Know

3. Have the authors made all data underlying the findings in their manuscript fully available?

Reviewer #1: No

Reviewer #2: No

Reviewer #3: Yes

4. Is the manuscript presented in an intelligible fashion and written in standard English?

Reviewer #1: Yes

Reviewer #2: No

Reviewer #3: No

5. Review Comments to the Author

Reviewer #1:

The present study investigated the biomechanical performance of various dental materials when filled into different cavity designs and their effects on surrounding dental tissues. The subject is intriguing and has the potential to contribute to the scientific literature.

Title:

Please indicate that this is an FEA study.

Response:

Thank for your comment. The title is revised as per your recommendation.

“Biomechanical performance of resin composite on dental tissue restoration: A finite element analysis”

Abstract:

Loading information is missing.

Response:

Thank you for your feedback; we have updated the abstract to include the occlusal load information as suggested.

“An occlusal load of 600 N was distributed on the top surface of the teeth to carry out simulations.”

Introduction:

The introduction lacks an explanation of different cavities and their significance as evaluation factors. Additional information about glass carbomer cement and its comparison with resin composite and amalgam should be introduced. Include a hypothesis at the end of this section.

Response:

Thanks for your comments to strengthen the introduction. The following paragraphs are added to the introduction as per your suggestion to strengthen the manuscript. 

“According to the G.V. Black classification system, a demineralization lesion situated within the pits and fissures of the occlusal (biting) surface is denoted as a Class I lesion, whereas a lesion found on the proximal surface of a posterior tooth is classified as a Class II lesion [5]. Within the Class II category, there exist various subtypes such as mesial and occlusal surfaces (MO) and mesial, occlusal, and distal surfaces (MOD) cavities [6].

Restorative dental materials commonly employed for direct placement, amalgam has traditionally held a prominent position, although concerns regarding its mercury content and non-aesthetic appearance have led to diminished usage in various communities [8]. In response to these concerns, novel resin-based composites have been developed, characterized by unique biomechanical properties such as cost-effectiveness, non-toxicity, prolonged durability, and enhanced aesthetic qualities [9].

Glass ionomer cements (GIC) are now used as both a restorative material and a fissure sealant in dentistry. It represents a relatively recent commercial development aimed at intentionally remineralizing damaged caries of enamel and dentin [10]. Recent advancements in material development have given rise to glass carbomer cement (GCC), an innovative variant of GICs, which incorporates nanoscale particles and introduces fluorapatite as an additional filler [11]. The flexural strength of GCC was found to be comparable to that of conventional glass ionomer cement [12], underscoring the advancements in material development. Despite these improvements, GICs are still not generally considered the top choice for restorative materials compared to amalgam and resin composites. Statistics show that composite materials (44%) are more commonly used than amalgam (40.9%), and glass-ionomers (13.4%) [10].

The first hypothesis posits that an infected tooth will have a significant difference in stress distribution in enamel and dentin compared to a restored or healthy tooth. The second hypothesis suggests that the stiffness of restorative materials significantly affects the stress distribution in restored teeth.”

Methods:

The tooth was modeled using a reference image and freeform features. Provide a description of the tooth number and shape. 

Response:

We have modeled the first molar (tooth number 30) located in the right mandible. The first molar has a typical molar shape characterized by multiple cusps and a broad occlusal surface. Figure 1 shows the shape and location of the modelled tooth.

“A first molar tooth (tooth number 30 in the right mandible) was modeled using a reference image and freeform feature. This tooth has a typical molar shape (a rectangular shape with pits and fissures on it) characterized by multiple cusps and a broad occlusal surface as shown in Fig 1.”

Explain how cortical bone thickness was defined. 

Response:

We established the cortical bone thickness by utilizing thickness features within the SolidWorks software. After designing the surfaces representing the cortical bone layer and the internal architecture of the cancellous bone core, we applied these thickness features to accurately model a cortical bone layer of 1.5 mm, effectively encircling the internal cancellous bone structure.

“The thickness of the cortical bone was modeled as 1.5 mm thick using a thickness feature within SolidWorks after designing surfaces to encompass the internal core of the cancellous bone.”

Justification for not including the PDL and pulp in the design needs revision. While it's acknowledged that soft tissue could affect stress distribution, the statement claiming no effect should be omitted

Response:

Thanks for highlighting the importance of modelling. It is important to emphasize that our decision aligns with established practices in similar studies. Many research articles in the field of dental biomechanics have consistently omitted the PDL, especially when focusing on stress distribution at the crown level rather than the root. This exclusion is substantiated by the well-documented characteristics of the PDL, including its low stiffness, with a frequently used value of E at 68.9 MPa, and its thin structure, typically measuring 0.2 mm in thickness. These attributes, supported by existing literature, suggest that the PDL's influence on peak stresses at the tooth crown level is negligible. Therefore, based on both precedent and the physical properties of the PDL, its exclusion from our design was a scientifically justified and the justification is added to the manuscript. 

https://doi.org/10.1016/j.jmbbm.2022.105305

https://doi.org/10.1080/01694243.2016.1195953

https://doi.org/10.1080/13102818.2015.1015445

https://doi.org/10.1080/13102818.2017.1373600

https://doi.org/10.1016/j.jmbbm.2021.104892

https://doi.org/10.1016/j.dental.2020.11.008

https://doi.org/10.1016/j.jds.2017.03.010

https://doi.org/10.1016/j.dental.2009.07.014

https://doi.org/10.1016/j.dental.2021.01.020

“The thickness of the cortical bone was modeled as 1.5 mm thick using a thickness feature within SolidWorks after designing surfaces to encompass the internal core of the cancellous bone. The PDL (periodontal ligament) and the pulp were not included in our design because it has been established in the literature that when simulating stress distribution at the crown level (regions of interest), the PDL is often excluded due to its well-documented characteristics, including its low stiffness (around 68.9 MPa) and its thin structure (typically 0.2 mm in thickness). These attributes suggest that the PDL's influence on peak stresses at the tooth crown level is negligible. Therefore, the exclusion of PDL and pulp modeling aligns with established modeling techniques [22].”

During the validation process, it was observed that the Guth model's prediction improved with increasing fiber volume fraction, whereas properties with the Counto model decreased. Provide a justification for this observation. 

Response:

We appreciate the reviewer's comment and would like to clarify that in our study, both the Guth and Counto models indeed exhibited an increase in Young's modulus with an increasing fiber volume fraction. The observed discrepancy between the two models arises from their distinct formulas for calculation. The Guth model inherently provides higher Young's modulus values compared to the Counto model due to the mathematical differences in their predictions. It's crucial to recognize that these models are based on unique theoretical foundations and, therefore, yield varying results. The consistent trend of Young's modulus increasing with fiber volume fraction in the Guth model not only supports the validity of our findings but also underscores its utility for comparative purposes. We specifically utilized the Guth model because it provides Young's modulus values at certain volume fractions that closely resemble those of amalgam materials, facilitating a meaningful and relevant comparison within the context of our study.

Clarify that Young's Modulus is theoretical. 

Response:

Thank you for your comment.

“Various theoretical models were employed to predict elastic moduli of micro-filled composite materials, which were compared with experimental results and showed good agreement [24]. Therefore, this study used these models (Eq 1 – 5) to predict the elastic moduli of composites.

Five Young’s moduli of the theoretically calculated composite (10, 14, 18, 22, and 26 GPa) were chosen for the simulation.”

Specify the aspect ratio of your mesh. 

The aspect ratio of the mesh was consistently maintained below 5. Thank you for your comment.

“The good quality of the mesh was ensured by keeping the aspect ratio below 5.”

Explain the choice of focusing on von-Mises stress over the first principal stress.

Response:

Thank you for your valuable comment. Von-Mises stress is a scalar quantity that takes into account the magnitude of stresses (considering both tensile and compressive stresses). Von-Mises stress provides a valuable measure of stress intensity that is particularly useful for assessing material failure, as it considers the combined effect of both the magnitude and the spread of stress in multiple directions, making it a more comprehensive indicator for structural analysis and material design. Moreover, von-Mises stress is commonly used in various biomechanical studies due to its applicability in characterizing stress states. However, it's important to note that we also assessed the first principal stress as part results of our study to provide a more comprehensive understanding of the stress distribution within the dental tissues.

Discussion:

Enhance the discussion regarding the comparison of different cavity designs. Provide more information about model validation and FEA limitations concerning PDL, soft tissue, and parafunctional loading.

Thank you for your comment. Some references were added in the discussion for the model validation including FE related research. We have incorporated FEA limitations at the end of the discussion as well.

This study examines three different restorative materials (Amlgam, GCC, Composite) in relation to three categories of dental cavities, specifically Class I and Class II cavities, which include MO and MOD types. Teeth lacking any form of restoration are designated as infected teeth, while teeth with restorations are categorized as restored teeth. In the field of restorative dentistry, understanding the behavior of restored teeth has proven to be a multifaceted challenge. This complexity arises from the wide array of restorative materials available, encompassing both traditional and advanced options as each material type can exhibit distinct properties and behaviors. These advancements have the potential to overcome the limitations of amalgam material, including their ability to control mineral loss and modulate biofilm formation. Regarding GIC, limited evidence [39] suggests that restorations made with this material exhibit a notably superior preventive effect against the development of new caries when compared to other restoration materials. 

The results revealed that the first hypothesis, which proposed that there would be a significant difference in stress distribution between infected, healthy, and restored teeth, was true. The findings indicated that stress levels were notably higher in infected teeth when compared to both restored and healthy teeth. This difference in stress distribution can be attributed to variations in the load application area within the cases under investigation. Specifically, infected teeth exhibited a reduced occlusal area subjected to a 600 N force in comparison to their healthy counterparts. It is important to note that the intensity of a distributed load is contingent upon the force per unit area. Therefore, teeth with cavities transmit a heightened level of stress to the dentin tissue. Consequently, the reduction in tooth structure significantly diminishes tooth stiffness [40]. The findings revealed that as more tooth material was removed, an increased concentration of stresses was observed within the enamel and dentin tissues. 

FEA stands as a potent tool in the realms of bioengineering and dentistry, and our study's primary objective was to scrutinize the repercussions of dental cavities and different restorative materials on the performance of enamel and dentin tissues.

It is essential, however, to acknowledge certain limitations inherent to our research. Notably, we employed a simplified assumption by considering all materials as homogenous, isotropic, and linearly elastic. Furthermore, our loading conditions were somewhat idealized, and variations such as oblique or parafunctional loading could impose varying stress patterns on the tooth structure. In this study, finite element models did not include PDL considering that the exclusion of PDL affects minimal to the stresses in the crown and surrounding tissues [22]. Another facet of consideration is the derivation of Young's modulus for the composite material, which was based on theoretical models rather than empirical testing. Although these models have been substantiated in previous research, the importance of experimental verification of the calculated Young's modulus cannot be overstated. Hence, prospective studies should prioritize the conduction of such experiments to corroborate the accuracy of the theoretical models employed here, thereby enriching our comprehension of composite material properties.

Conclusion:

The conclusion is overly lengthy. Condense the text, remove redundant results, and present only the most clinically significant findings to the readers.

Response:

The conclusion is revised and key finding are listed in the conclusion. We believe that now conclusion is comprehensive and in a better shape.

From this study, it was concluded that the geometry of cavities and different restorative materials affect the biomechanical performance of the restored tooth. Nanocomposite resin is a good alternative material for dental hard tissue restoration especially for patients who are concerned about the potential toxicity of mercury or aesthetic restoration. 

• Infected or damaged teeth with cavities are prone to higher stresses as compared to healthy teeth. 

• Larger cavities generate higher stresses in enamel and dentin tissues, thus, to protect the remaining teeth, the larger and smaller cavities should be filled as early as possible. 

• Without filling the cavity, the stresses may reach higher than the strength of the enamel which may cause further damage to the tooth.

• The restoration of Class II MOD cavity is very tricky and complex because of the removal of large tissue, which may require more care and maintenance during their normal function. 

• The selection of restorative material is the key to efficiently restoring the function of the tooth. Therefore, resin-based composites are good alternative materials due to the tailorable Young’s modulus according to the biomechanical and clinical requirements. 

• Overall, amalgam and composite C26 (26 GPa) outperformed biomechanically but composites are preferred over amalgam due to aesthetic and clinical concerns of amalgam.

 Reviewer #2:

Reviewer #2: Intro

-"Several factors lead to an ecological imbalance between (...) that contribute to creating tooth cavities [2]." Please add some short thoughts on the pre-cavity stage. Remember that this would seem even more (or at least equally) important to prevent progress of caries (and cavitation). Refer to https://www.quintessence-publishing.com/deu/de/article/840164/quintessence-international/2009/08/closing-the-gap-between-oral-hygiene-and-minimally-invasive-dentistry-a-review-on-the-resin-infiltration-technique-of-incipient-proximal-enamel-lesions, and to https://journals.plos.org/plosone/article?id=10.1371/journal.pone.0228249. Discuss adequately.

Response: 

Thanks for your comment. The introduction is revised and the literature from the mentioned papers has been added and cited as per your suggestion.

“However, modern dentistry prioritizes minimal intervention by emphasizing preventative and non-surgical approaches to safeguard tooth hard tissues before the formation of dental cavities. In this regard, the resin infiltration technique is widely used which involves the application of a low-viscosity resin to infiltrate porous and demineralized inter-crystalline spaces within enamel lesions [3]. In an ex vivo study [4], researchers investigated the treatment of non-cavitated and micro-cavitated caries using a modified resin incorporating silver nanoparticles (AgNP) for both internal and external infiltration, aiming to confer antimicrobial properties. The findings revealed that the AgNP-infused resin effectively hindered the regrowth of microbial biofilms while preserving the mechanical integrity of the enamel and preventing the recurrence of caries [3]. Timely intervention to address demineralized and porous enamel lesions is critical, as untreated lesions may progress to dental cavities.”

- Provide digestible units/divide into paragraphs.

Response:

Thank you for your comment. The introduction has been revised according to your suggestion and divided into paragraphs.

- Avoid Authors' names with your full text.

Response:

Thanks for your comment. The names of authors were removed from the text as per your suggestion.

- "The purpose of this study is to investigate (...)." Your study has already been finished, right? Must read "The purpose of this study was to investigate (...)."

Response: 

Thank you for pointing out the error; we have now corrected the sentence to read. 

"The purpose of this study was to investigate the behavior of dental tissues using alternative restorative materials to fill cavities of varying designs."

Results

- "(...) with cavity class I." must read "(...) with cavity Class I."

- Same with "cavity class II". Revise thoroughly.

Response: 

Thank you for your feedback; we have revised the manuscript to capitalize "cavity Class.. " as suggested.

Discussion

"The behavior of restored teeth is not well characterized due to the variety of restorative materials (both traditional and advanced materials)." Meaning would seem unclear, please expand your thoughts, and provide references.

Response: 

Thank you for your remarkable note. We have revised the discussion section, by adding following paragraphs:

“In the field of restorative dentistry, understanding the behavior of restored teeth has proven to be a multifaceted challenge. This complexity arises from the wide array of restorative materials available, encompassing both traditional and advanced options as each material type can exhibit distinct properties and behaviors. These advancements have the potential to overcome the limitations of amalgam material, including their ability to control mineral loss and modulate biofilm formation. Regarding GIC, limited evidence [39] suggests that restorations made with this material exhibit a notably superior preventive effect against the development of new caries when compared to other restoration materials.” 

- Same with "the stress was greater in the infected teeth". Please clarify.

- Refer to https://journals.sagepub.com/doi/10.1177/00220345890680111401

when explaining your thoughts.

Response:

Thank you for your valuable feedback. We have taken your suggestion into consideration and have now added a clarification regarding the higher stress observed in infected teeth. Additionally, we have included the reference you provided.

 The findings indicated that stress levels were notably higher in infected teeth when compared to both restored and healthy teeth. This difference in stress distribution can be attributed to variations in the load application area within the cases under investigation. Specifically, infected teeth exhibited a reduced occlusal area subjected to a 600 N force in comparison to their healthy counterparts. It is important to note that the intensity of a distributed load is contingent upon the force per unit area. Therefore, teeth with cavities transmit a heightened level of stress to the dentin tissue. Consequently, the reduction in tooth structure significantly diminishes tooth stiffness [40]. The findings revealed that as more tooth material was removed, an increased concentration of stresses was observed within the enamel and dentin tissues. 

- Again, provide digestible units by paragraphing.

Response:

Thank you for your suggestion. The discussion has been revised.

- Add some thoughts on the limitations of your study.

Thank you for your comment. We have added some limitations at the end of discussion section.

“It is essential, however, to acknowledge certain limitations inherent to our research. Notably, we employed a simplified assumption by considering all materials as homogenous, isotropic, and linearly elastic. Furthermore, our loading conditions were somewhat idealized, and variations such as oblique or parafunctional loading could impose varying stress patterns on the tooth structure. In this study, finite element models did not include PDL considering that the exclusion of PDL affects minimal to the stresses in the crown and surrounding tissues [22]. Another facet of consideration is the derivation of Young's modulus for the composite material, which was based on theoretical models rather than empirical testing. Although these models have been substantiated in previous research, the importance of experimental verification of the calculated Young's modulus cannot be overstated. Hence, prospective studies should prioritize the conduction of such experiments to corroborate the accuracy of the theoretical models employed here, thereby enriching our comprehension of composite material properties.”

Response:

Conclusion

- "The aim of this study was to analyze how (...)." This is not considered a conclusion from your study.

- Same with "Five restorative materials were studied, glass-carbomer cement, amalgam, and dental composite with stiffness variation (10 G Pa to 26 G Pa with an increment of 4)."

- Same with "Nanocomposite resin may be an excellent choice (...)."

- With your Conclusion section, please stick exclusively to your revised aims. Do not simply repeat your results here. Do not speculate. Instead, provide a reasonable and generalizable extension of your outcome.

Response: Thank you for your continued guidance and feedback on our conclusion section. We have carefully revised it to ensure that it aligns with your recommendations:

From this study, it was concluded that the geometry of cavities and different restorative materials affect the biomechanical performance of the restored tooth. Nanocomposite resin is a good alternative material for dental hard tissue restoration especially for patients who are concerned about the potential toxicity of mercury or aesthetic restoration. 

• Infected or damaged teeth with cavities are prone to higher stresses as compared to healthy teeth. 

• Larger cavities generate higher stresses in enamel and dentin tissues, thus, to protect the remaining teeth, the larger and smaller cavities should be filled as early as possible. 

• Without filling the cavity, the stresses may reach higher than the strength of the enamel which may cause further damage to the tooth.

• The restoration of Class II MOD cavity is very tricky and complex because of the removal of large tissue, which may require more care and maintenance during their normal function. 

• The selection of restorative material is the key to efficiently restoring the function of the tooth. Therefore, resin-based composites are good alternative materials due to the tailorable Young’s modulus according to the biomechanical and clinical requirements. 

• Overall, amalgam and composite C26 (26 GPa) outperformed biomechanically but composites are preferred over amalgam due to aesthetic and clinical concerns of amalgam.

We hope these revisions better address your concerns and appreciate your diligence in helping us improve the focus and clarity of our conclusion.

Refs

- Revise for uniform Journal style. No capital letters with the various Titles.

Response: 

Thank you for pointing out the need for uniform journal style with no capital letters in various titles.

In total, this would seem an interesting draft, easily intelligible, and considered ready to proceed after some revisions, please see below.

Response: 

Thank you for your positive assessment of the draft. We appreciate your feedback. We have already made the necessary revisions to the manuscript based on your suggestions.

Reviewer #3:

Reviewer #3: The goal of this study is to examine how dental tissues react when cavities of various shapes are filled with alternative restorative materials. Three classes of cavities were chosen, class I (occlusal), class II MO (mesio-occlusal), class II MOD (mesio-occlusal-distal), and three types of restoration material were selected, amalgam, glass carbomer cement (GCC), and silica-reinforced polymer composites with five different Young’s moduli. While it is good that the results have practical impact in terms of application, there are problems in modeling, especially in the creation of finite element models. Therefore, I think it needs to be revised and needs a second check.

Response: 

We appreciate your comment and checked other studies similar to this study and found that the models are correct based on reasonable assumptions. Besides that, the authors possess well established techniques in finite element modelling and analysis of implants. 

The figures and pictures are so blurry that it is difficult to read. Please enrich the quality of the figures and pictures.

Response: 

Your comment is greatly valued. The system of PLOSOne journal reduces the quality of the figures on the website, which is indicated during the submission process, however, high resolution figures can be download to the given link on the top right side of each figure. Below is the example of download and submitted high-resolution figure. 

System generated PDF file figure.

Submitted and downloaded high quality figure from right side link.

Wouldn't the periodental ligament have an effect on stress changes? Not considered in the modeling.

Response: 

Thank you for your thoughtful comment. It's important to note that many similar studies have also excluded the PDL in their simulations (see following articles), particularly when the cavities are located at the crown rather than the root. Additionally, the PDL's low stiffness (the most frequently used value of E is 68.9 MPa) and thin structure (0.2 mm thickness) make it unlikely to significantly impact peak stresses at the tooth crown level, as supported by the literature. 

https://doi.org/10.1080/01694243.2016.1195953

https://doi.org/10.1080/13102818.2015.1015445

https://doi.org/10.1080/13102818.2017.1373600

https://doi.org/10.1016/j.jmbbm.2021.104892

https://doi.org/10.1016/j.dental.2020.11.008

https://doi.org/10.1016/j.jds.2017.03.010

https://doi.org/10.1016/j.jmbbm.2022.105305

https://doi.org/10.1016/j.dental.2009.07.014

https://doi.org/10.1016/j.dental.2021.01.020

The following text is added to the manuscript for clarification.

“The PDL (periodontal ligament) and the pulp were not included in our design because it has been established in the literature that when simulating stress distribution at the crown level (regions of interest), the PDL is often excluded due to its well-documented characteristics, including its low stiffness (around 68.9 MPa) and its thin structure (typically 0.2 mm in thickness). These attributes suggest that the PDL's influence on peak stresses at the tooth crown level is negligible. Therefore, the exclusion of PDL and pulp modeling aligns with established modeling techniques [22]”

The Author said that ”Each 3D model of the tooth was meshed by finer linear tetrahedral elements (size of 0.05 for the tooth), and the bone was meshed using coarse elements to reduce computational cost”. However, while finite elements are a way of representing the structure as an elastic structure, the element quality has a very important effect on this representation. Three classes of cavities were used. However, there will be geometry changes. How did the same element size suffice for all of them? Linear element was used and compared to quadratic type, tetrahedron element has poor performance and high margin of error. I suggest the authors to do a convergence analysis and quadratic element.

Response: 

Thank you for your valuable comment. We acknowledge the importance of element quality in finite element analysis and appreciate your suggestion regarding a convergence analysis using quadratic elements. In our study, we maintained a consistent element size for all models to ensure uniform meshing conditions. We appreciate your suggestion and will consider it for future work to further refine our modeling and analysis. It's worth mentioning that the authors possess well established techniques in finite element modelling and analysis of implants including the advantages of quadratic elements for structures with complex geometries like bones and teeth. However, in this particular study, mesh convergence was rigorously conducted using both linear tetrahedral and quadratic elements with the same seed size to ensure consistency across all models. Interestingly, the results from both element types yielded similar outcomes. This suggests that, for the specific focus of this study, the linear tetrahedral elements were adequate in capturing the relative differences in biomechanical performance among the different cavity classes.

It would be better for comparison if the fringes used in the figures in the stress distributions have the same range.

Response: 

Thank you for your suggestion. While using the same legend for all stress distributions may aid in direct comparison, it's important to note that it can make it challenging to discern peak stresses, particularly in regions with small elements. To maintain clarity and readability in our figures, we opted to use legends with different ranges for stress distributions, which allows us to emphasize critical stress variations and make them more apparent to the reader.

I recommend the author to add the following publication to the literature search. In this publication "The restorative material type changed the stress distribution of endocrown restorations. The fiber application did not affect the stress distribution in either endocrown group. But, more stress absorption was observed in fiber under IPS-E than C-E." is the conclusion that will contribute to the content of the publication.

Response: 

Thank you for your valuable suggestion, and we appreciate your input. We have now included the recommended publication in our literature search to enhance the content of our manuscript.

“In conjunction with the aforementioned findings regarding the influence of different cavities and restorative materials on stress distribution in teeth, a study [47] explored the impact of restorative material choice on stress distribution within endocrown restorations. The research revealed that the selection of restorative material had a notable effect on stress distribution patterns within endocrown restorations, while the application of fibers did not significantly alter stress distribution in either endocrown group.”

6. PLOS authors have the option to publish the peer review history of their article (what does this mean?). If published, this will include your full peer review and any attached files.

Do you want your identity to be public for this peer review? For information about this choice, including consent withdrawal, please see our Privacy Policy.

Reviewer #1: No

Reviewer #2: No

Reviewer #3: No

---

## [Decision Letter · Decision Letter 1]

7 Nov 2023

PONE-D-23-26356R1Biomechanical performance of resin composite on dental tissue restoration: A finite element analysisPLOS ONE

Dear Dr. Mehboob,

Thank you for submitting your manuscript to PLOS ONE. After careful consideration, we feel that it has merit but does not fully meet PLOS ONE’s publication criteria as it currently stands. Therefore, we invite you to submit a revised version of the manuscript that addresses the points raised during the review process.

We look forward to receiving your revised manuscript.

Kind regards,

Artak Heboyan, Ph.D.

Academic Editor

PLOS ONE

Journal Requirements:

Reviewers' comments:

Reviewer's Responses to Questions

**Comments to the Author**

1. If the authors have adequately addressed your comments raised in a previous round of review and you feel that this manuscript is now acceptable for publication, you may indicate that here to bypass the “Comments to the Author” section, enter your conflict of interest statement in the “Confidential to Editor” section, and submit your "Accept" recommendation.

Reviewer #1: All comments have been addressed

Reviewer #2: All comments have been addressed

Reviewer #3: All comments have been addressed

2. Is the manuscript technically sound, and do the data support the conclusions?

Reviewer #1: Yes

Reviewer #2: No

Reviewer #3: Yes

3. Has the statistical analysis been performed appropriately and rigorously? 

Reviewer #1: N/A

Reviewer #2: Yes

Reviewer #3: Yes

4. Have the authors made all data underlying the findings in their manuscript fully available?

Reviewer #1: Yes

Reviewer #2: Yes

Reviewer #3: Yes

5. Is the manuscript presented in an intelligible fashion and written in standard English?

Reviewer #1: Yes

Reviewer #2: Yes

Reviewer #3: Yes

6. Review Comments to the Author

Reviewer #1: I am thoroughly satisfied with the current version as it fulfills all my requirements and expectations to the fullest extent of this paper.

Reviewer #2: With the help of the reviewers, this revised and re-submitted manuscript has been considerably improved. Notwithstanding, still some minor issues would seem in need of revision, please see below.

- "The first hypothesis posits that (...)" and "The second hypothesis suggests that (...)" - please provide a sound and valid NULL HYPOTHESIS. The latter must be deducible from the forgoing thoughts.

- "The results revealed that the first hypothesis, which proposed that there would be a significant difference in stress distribution between infected, healthy, and restored teeth, was true." Please note that a hypothesis never will be "true". Again, please provide a sound NULL hypothesis (see comments given above), and remember that H0 can be rejeceted or not rejected.

- Same with "The second hypothesis of the study (...) was also true." Revise carefully.

- "This highlights the importance of early detection and treatment of dental infections to prevent the progression of damage and the need for extensive restorative procedures." Sound references missing. Again, please stick to https://pubmed.ncbi.nlm.nih.gov/19639091/, to https://doi.org/10.13140/RG.2.2.36646.37443, and to https://pubmed.ncbi.nlm.nih.gov/28294198/. Please discuss.

- "An in vitro study reported that the use of composite resin reduces the amount of stress concentration at the interface between the adhesive and the tooth in comparison with other materials that have a higher Young modulus than enamel [42]." Remember that one sentence does not constitute one paragraph. Please provide digestible units.

This re-submitted manuscript will be worth following after some further revisions.

Reviewer #3: Although there are points about the number of elements, convergence study and quadratic element that I disagree with the author, I think that it is acceptable due to relative comparisons within itself. Other explanations and arrangements are sufficient and I think that the article can be published in your journal.

7. PLOS authors have the option to publish the peer review history of their article (what does this mean?). If published, this will include your full peer review and any attached files.

Reviewer #1: No

Reviewer #2: No

Reviewer #3: No

---

## [Author Response · Author response to Decision Letter 1]

17 Nov 2023

Reviewer #1:

I am thoroughly satisfied with the current version as it fulfills all my requirements and expectations to the fullest extent of this paper.

Response: 

Thank you for accepting the revised version; I appreciate your approval for the article's publication in the journal.

 Reviewer #2:

With the help of the reviewers, this revised and re-submitted manuscript has been considerably improved. Notwithstanding, still some minor issues would seem in need of revision, please see below.

1- “The first hypothesis posits that (...)" and "The second hypothesis suggests that (...)" - please provide a sound and valid NULL HYPOTHESIS. The latter must be deducible from the forgoing thoughts.

Response:

Thank you for your valuable suggestions to enhance the manuscript. The introduction has been revised incorporating the following paragraph: 

“The first null hypothesis maintains that no notable disparity exists in stress distribution within enamel and dentin between an infected tooth and a restored or healthy tooth. The second null hypothesis contends that the stiffness of restorative materials holds no consequential influence on the stress distribution observed in restored teeth.”

2- “The results revealed that the first hypothesis, which proposed that there would be a significant difference in stress distribution between infected, healthy, and restored teeth, was true." Please note that a hypothesis never will be "true". Again, please provide a sound NULL hypothesis (see comments given above), and remember that H0 can be rejected or not rejected.

Response:

Thanks for your comments to improve the manuscript. We have revised carefully as suggested by reviewer.

The results revealed that the first null hypothesis, suggesting no significant difference in stress distribution between infected, healthy, and restored teeth, was not supported and was consequently rejected.

3- Same with "The second hypothesis of the study (...) was also true." Revise carefully.

Response:

The study's findings led to the rejection of both null hypotheses. Thank you for your valuable insights.

“The second null hypothesis, which stated that the stiffness of restorative materials has no consequential influence on the stress distribution observed in restored teeth, was also not supported and therefore was rejected.”

4- "This highlights the importance of early detection and treatment of dental infections to prevent the progression of damage and the need for extensive restorative procedures." Sound references missing. Again, please stick to https://pubmed.ncbi.nlm.nih.gov/19639091/, to https://doi.org/10.13140/RG.2.2.36646.37443, and to https://pubmed.ncbi.nlm.nih.gov/28294198/. Please discuss.

Response:

We appreciate your valuable input in enhancing the manuscript. Following your guidance, the references mentioned above have been incorporated to strengthen and improve the manuscript.

“Addressing early detection and treatment of dental infections promptly becomes pivotal in preventing the progression of damage and the subsequent necessity for extensive restorative techniques. Research investigating resin infiltration techniques [3,43,44] substantiates this notion, demonstrating their capacity to halt the advancement of white spot lesions. This underscores the importance of timely intervention as a preventive measure against the need for extensive dental restorations”.

3. Kielbassa AM, Muller J, Gernhardt CR. Closing the gap between oral hygiene and minimally invasive dentistry: a review on the resin infiltration technique of incipient (proximal) enamel lesions. Quintessence Int. 2009;40: 663–81.

43. Kielbassa AM, Ulrich I, Werth VD, Schüller C, Frank W, Schmidl R. External and internal resin infiltration of natural proximal subsurface caries lesions: A valuable enhancement of the internal tunnel restoration. Quintessence Int (Berl). 2017;48.

44. Kielbassa AM, Ulrich I, Treven L, Mueller J. An updated review on the resin infiltration technique of incipient proximal enamel lesions. Med Evol. 2010;16: 3–15.

- "An in vitro study reported that the use of composite resin reduces the amount of stress concentration at the interface between the adhesive and the tooth in comparison with other materials that have a higher Young modulus than enamel [42]." Remember that one sentence does not constitute one paragraph. Please provide digestible units.

Response:

We appreciate your guidance to enhance the manuscript; the paragraph below has been revised accordingly

“Composite resins, designed to replicate the natural appearance of teeth and closely mimic their mechanical properties, demonstrate a favorable attribute in adapting to the natural flexure and movement of teeth during function. This adaptability potentially contributes to a more uniform distribution of stress forces, mitigating the likelihood of stress concentration points at the adhesive-tooth interface. An in vitro study corroborated this behavior, demonstrating that the use of composite resin reduces stress concentration at the interface between the adhesive and the tooth compared to materials with a higher Young's modulus than enamel [42].”

Reviewer#3:

Reviewer #3: Although there are points about the number of elements, convergence study and quadratic element that I disagree with the author, I think that it is acceptable due to relative comparisons within itself. Other explanations and arrangements are sufficient and I think that the article can be published in your journal.

Response: 

Thank you for approving the revisions and for your recommendation to publish the article in the journal.

The authors thank all the reviewers for their valuable comments to improve the paper quality.

---

## [Decision Letter · Decision Letter 2]

23 Nov 2023

Biomechanical performance of resin composite on dental tissue restoration: A finite element analysis

PONE-D-23-26356R2

Dear Dr. Mehboob,

We’re pleased to inform you that your manuscript has been judged scientifically suitable for publication and will be formally accepted for publication once it meets all outstanding technical requirements.

Kind regards,

Artak Heboyan, Ph.D.

Academic Editor

PLOS ONE

Additional Editor Comments (optional):

Reviewers' comments:

Reviewer's Responses to Questions

**Comments to the Author**

1. If the authors have adequately addressed your comments raised in a previous round of review and you feel that this manuscript is now acceptable for publication, you may indicate that here to bypass the “Comments to the Author” section, enter your conflict of interest statement in the “Confidential to Editor” section, and submit your "Accept" recommendation.

Reviewer #2: All comments have been addressed

2. Is the manuscript technically sound, and do the data support the conclusions?

Reviewer #2: Yes

3. Has the statistical analysis been performed appropriately and rigorously? 

Reviewer #2: Yes

4. Have the authors made all data underlying the findings in their manuscript fully available?

Reviewer #2: Yes

5. Is the manuscript presented in an intelligible fashion and written in standard English?

Reviewer #2: Yes

6. Review Comments to the Author

Reviewer #2: This revised and and re-submitted manuscript is considered ready to proceed. Congrats, and stay healthy!

7. PLOS authors have the option to publish the peer review history of their article (what does this mean?). If published, this will include your full peer review and any attached files.

Reviewer #2: No

---

## [Editor Report · Acceptance letter]

12 Dec 2023

PONE-D-23-26356R2 

Biomechanical performance of resin composite on dental tissue restoration: A finite element analysis 

Dear Dr. Mehboob:

I'm pleased to inform you that your manuscript has been deemed suitable for publication in PLOS ONE. Congratulations! Your manuscript is now with our production department. 

Kind regards, 

on behalf of

Dr. Artak Heboyan 

Academic Editor

PLOS ONE